# Spatio-Temporal Changes of Extracellular Matrix (ECM) Stiffness in the Development of the Leech *Hirudo verbana*

**DOI:** 10.3390/ijms232415953

**Published:** 2022-12-15

**Authors:** Laura Pulze, Nicolò Baranzini, Terenzio Congiu, Francesco Acquati, Annalisa Grimaldi

**Affiliations:** 1Department of Biotechnology and Life Sciences, University of Insubria, Via J.H. Dunant 3, 21100 Varese, Italy; 2Department of Medical Sciences and Public Health, University of Cagliari, Cittadella Universitaria, 09042 Monserrato, CA, Italy

**Keywords:** microenvironment, extracellular matrix, stiffness, leech, development, collagen

## Abstract

The invertebrate leech *Hirudo verbana* represents a powerful experimental animal model for improving the knowledge about the functional interaction between the extracellular matrix (ECM) and cells within the tissue microenvironment (TME), and the key role played by ECM stiffness during development and growth. Indeed, the medicinal leech is characterized by a simple anatomical organization reproducing many aspects of the basic biological processes of vertebrates and in which a rapid spatiotemporal development is well established and easily assessed. Our results show that ECM structural organization, as well as the amount of fibrillar and non-fibrillar collagen are deeply different from hatching leeches to adult ones. In addition, the changes in ECM remodelling occurring during the different leech developmental stages, leads to a gradient of stiffness regulating both the path of migratory cells and their fates. The ability of cells to perceive and respond to changes in ECM composition and mechanics strictly depend on nuclear or cytoplasmic expression of Yes-Associated Protein 1 (YAP1), a key mediator converting mechanical signals into transcriptional outputs, expression, and activation.

## 1. Introduction

The extracellular matrix (ECM) is a highly specialized three-dimensional network composed of a wide range of molecules such as collagens, fibronectin, laminins and proteoglycans (PGs), the last composed of a core protein and GAG chains, in particular hyaluronan (HA) [1,2,3,4]. The complexity of such a molecular milieu is further increased by the presence of different cell types, such as endothelial cells and their precursors, muscle cells, adipocytes, fibroblasts and a vast representation of cells belonging to the immune system. Altogether, this combination of molecular and cellular components gives rise to an extremely dynamic and complex biological entity, called the tissue microenvironment (TME).

It has long been documented that the ECM provides support and anchorage to nearby cells, maintaining their physical structure and dictating their functional properties [4]. Indeed, the reciprocal crosstalk between the ECM and the cellular components of the TME shows a dynamic interplay, which is mainly (but not exclusively) mediated by a complex repertoire of cell surface receptors, such as integrins, growth factor receptors, cytokine/chemokine receptors, Toll-like receptors (TLRs) and their corresponding ligands [5,6,7,8,9]. The underlying chemical and mechanical signals provided by the ECM are in turn involved in the regulation of an impressive set of key biological processes, among which intercellular communication, cell survival and proliferation, migration/invasion pattern and differentiation stand out [10,11,12,13,14].

As expected, the dynamic cellular and molecular crosstalk within the TME entails a significant ECM turnover and remodelling, which occurs under both physiological and pathological conditions, such as inflammation and wound healing [15,16,17,18].

Despite a plethora of published data on the multifaceted role of the ECM in animal homeostasis, a full understanding of how spatiotemporal changes of ECM components and their structure affect cell behaviour is still lacking. In this context, the availability of a simple and cost-effective experimental animal model whose morphological and physiological features are well established is of paramount relevance.

The medicinal leech has recently emerged as a valuable invertebrate animal model for dissecting the cellular and molecular mechanisms involved in embryo development, immune system responses, wound healing, angiogenesis, and fibroplasia, because of its unique features and evolutionarily conserved mechanisms with mammals [19,20,21,22,23,24,25,26,27,28].

The anatomical organization of this invertebrate is relatively simple, characterized by a soft muscular body and reduced cavities, and reproduces many aspects of vertebrate basic biological processes [19,20]. The body wall is made predominantly of muscle fibres organized in fields and contains different organs.

In this paper, we shed light on how fine tuning of the 3D ECM microenvironment can drive both cellular and tissue fates and their spatial organization; moreover, we describe how a particular tissue adapts its inner organization in response to the physical and chemical architecture of the ECM microenvironment; and finally, how different cell types integrate the cues related to mechanical elasticity or stiffness of the ECM in a timely manner to determine their polarity and differentiation pattern.

## 2. Results

The body of the adult *Hirudo verbana* (7–8 cm in length and about 1 cm wide) consists of a muscular-cutaneous sac of about 800–1000 μm diameter in cross sectioned leeches. Between the body wall and the gut, the loose connective tissue is populated by different tissue-resident, widely spread cells represented in limited numbers: macrophages, granulocytes, NK cells, telocytes, botryoidal cells, vasocentral/vasofibrous cells, and fibroblasts (the latter being the predominant source of ECM components).

Under the epithelium, muscle fibres belonging to the helical muscle type, represent the predominant cell type and are organized in multiple layers disposed circularly, obliquely and longitudinally.

The longitudinal muscle fibres were found to be organized in discrete sectors separated by a scant connective tissue and, occasionally, by lateral dorso-ventral muscles, as observed in cross-sectioned body wall (Figure 1A).

These helical muscle fibres (mononucleated muscle cells described by Lanzavecchia et al.) [29] form groups of about 10/15 elements, lined by thick connective-tissue capsules.

Accordingly, both Masson’s trichrome and Sirius red staining of leech sections from adults revealed a modest amount of collagen fibres staining blue (Figure 1B) and red, respectively (Figure 1C).

Fifty day-old juvenile leeches (2 cm in length and about 0.5 cm wide) can swim actively and move with compass-like movements. Their body wall showed the same gross arrangement observed in adult individuals, but the connective tissue displayed a different spatial distribution, whereby the muscular-cutaneous sac (about 250–300 μm in cross sectioned animals) was mainly constituted of small muscle fibre groups, surrounded by abundant connective tissue (Figure 1D). The total collage distribution, as detected by Masson’s trichrome and Sirius red, was thick and compact (Figure 1E,F).

When hatched from the cocoon, leeches are very small (0.5 cm in length and about 3 mm wide). New-born leeches prefer to live attached to a substrate but are nevertheless mobile and able to swim—just as juvenile and adult animals—despite the structural organization of their body wall, which is significantly different. In fact, unlike juveniles and adults, the new-born’s connective tissue under the cuticle and the epithelium is more abundant: the dispersed muscle fibres are small and wrapped by more or less stiffened collagen (Figure 1G), as revealed by Masson’s trichrome and Sirius red staining (Figure 1H,I). At this developmental stage several cells, very small in size, were observed in the parenchyma between the body wall (about 100–150 μm in cross sectioned leeches) and the gut (Figure 1G).

### 2.1. TEM and SEM Analysis

Ultrastructural analyses have been performed to shed light on the substantial differences among adult, juvenile and hatching leeches, and in ECM abundance and stiffness, mainly due to the supramolecular organization of collagen molecules, as highlighted in histological and histochemical samples.

In the adult *Hirudo verbana*, the ECM in the body wall was under-represented and collagen fibres formed a “basket” around each group of muscle cells (Figure 2A–D). In each group, adjacent fibres characterized by complementary surfaces were completely separated from their neighbours by a scarce ECM and only collagen structures linked the clustered fibres with the capsule (Figure 2E). Each muscle was thus represented by a mononucleated round circomyarian fibre showing, in cross section, the contractile material forming a thick continuous ring around the cytoplasmic axis filled with cellular organelles (Figure 2C,F). In cross section, the mature muscle fibres were characterized by up to 15 series of myosin/actin filaments, from the surface to the centre, organized in “sarcomere”-like structures defined by Z elements (Figure 2F).

In 50-day old juvenile leeches, by contrast, the ECM architecture was different from that observed in the adult tissue stroma by showing (in a radial direction from epithelium toward the gut) a complex gradient in density, composition and architecture (Figure 3A–D). In the innermost part of the body, thick bundles of collagen, oriented parallel and orthogonally with respect to the main body axis, were alternated with areas characterized by loose connective tissue (Figure 3B).

Close to the body surface, the ECM was less organized and the few collagen fibres surrounded small groups of muscles. Matrix stiffening was apparently induced by many activated fibroblasts involved in collagen deposition and packaging and also by ECM cross-linking. Of note, the ECM assembly process (from which depends both the concentration gradient of deposition and the physical ECM properties—such as stiffness and molecular composition) was mirrored by the differentiated state of the cells forming the body wall.

Still closer to the body surface, a few muscles, small in size but organized as round cyrcomiarian muscle fibres, were grouped and delimited by fibrillar organized collagen (Figure 3E). In both cross and radial direction, five to six series of myosin and actin filaments, organized in “sarcomeres” already defined by short Z elements, were evident (Figure 3F).

In between the most external and most internal zones of the leech body, in the corridors delimited by thick bundles of collagen, myocytes displaying a migrating phenotype were surrounded only by an ECM with loose properties (Figure 3G) while, closer to the gut, small myocytes characterized by large nuclei and very few filaments appeared scattered under the sarcolemma (Figure 3H,I).

Interestingly, recently hatched leeches show a peculiar organization of ECM with respect to its distribution in juvenile and adult body wall. The fibril and non-fibril components of ECM were predominant compared to their cellular counterparts. The wide collagen scaffold, coupled to an extensive collagen cross-linking, greatly stiffened the tissue stroma. Collagen fibrils were organized in radially arranged thick bundles, to mark the limits of the areas characterized by a less dense ECM network (Figure 4A–E). This compartmentalization imposes a clear limitation on the path that cells (mostly myocytes) could take to migrate from the centre to the periphery in order to reach their final position. The migrating phenotype of these myocytes is confirmed by their elongated and ruffled profile, the presence of focal adhesions scattered along the sarcolemma and the dense basal membrane surrounding each cell (Figure 4G,H). These very small cells were found in an early differentiation state, as demonstrated by the reduced peripheral contractile layer (Figure 4F).

Just like muscular cells, immunocytes and interstitial cells are also produced in the central zone close to the gut and then migrate along the channels delimited by collagen toward the superficial area (Figure 4E).

### 2.2. Characterization of ECM Components

In leeches, three types of collagens are detected by immunofluorescence and immunoblotting according to the different developmental stages.

Collagens are usually categorized as either fibrillar or non-fibrillar. Collagen I and III (fibrillar types) were co-expressed starting from the newly born to the adult worm stage. Even if collagen I always represented the main component (Figure 5A,D,G) collagen III was present in larger quantity in hatched and juvenile leeches compared to adults. Collagen III was thus co-expressed with type I, but its amount decreased with ongoing development (Figure 5B,E,H). By contrast, non-fibrillar collagen IV (Figure 5C,F,I), the major constituent of basement membrane, was highly expressed, mainly during the first steps of development, peaking in recently hatched leeches (Figure 5I). All the immunofluorescence data were confirmed by immunoblot (Figure 6A–C).

Proteoglycans (PGs) represent a core constituent of ECM. They are composed of a core protein and GAG chains, in particular hyaluronan (HA). HA can form aggregates that interacts with other ECM components (Figure 7) to provide tissue resilience.

Concerning the distribution pattern of PGs and HA (assessed by conventional histochemical procedures and immunofluorescence assays, respectively), the amount of both types of molecule were abundant in newly hatched leeches, to later decrease mildly in juvenile and drastically in adults (Figure 7A,C,D,H,G,K). In addition, PGs were visualised as electron dense filaments closely associated with the collagen fibrils (Figure 7B,E,I,J).

### 2.3. YAP1 Detection

The ECM stiffness regulates the ability of cells to develop forces through their contractile cytoskeleton and to assemble focal adhesions. Many studies have unveiled YAP1 as a key mediator that converts mechanical signals into transcriptional outputs; in particular, YAP1 subcellular localization is modulated upon exposure to different mechanic stimuli, such as ECM stiffness and forces from nearby cells, inducing cytoskeletal rearrangement [30].

In order to evaluate the correlation between ECM stiffness and YAP1 expression, we first performed a bioinformatic analysis to evaluate the YAP1 degree of conservation. The amino acid sequence of *H. verbana* YAP1 was aligned with those from *H. vulgaris*, *T. spiralis* and *M. galloprovincialis* (Figure 8A), highlighting the high conservation of a WW motif also in the medicinal leech protein. A similar result was also obtained by comparing human and leech YAP1, revealing a fundamental similarity with vertebrates as well (Figure 8B).

Accordingly, Western blot analysis showed that a pick of total YAP1 protein expression (i.e., the protein present both in nuclei and cytoplasm) was observed in juvenile leeches, but it turned out to be drastically reduced in adult and recently hatched leeches (Figure 9A). The specific tissue localization of YAP1 was assessed by immunocytochemical staining, showing that in hatched and juvenile leeches YAP1 was preferentially localized in the nucleus (Figure 9D–G), while in adults it was more evident in the cytoplasm of body wall cells (Figure 9B,C).

## 3. Discussion

Our state of knowledge about the functional interplay between ECM and cells within the TME and, more specifically, the key role played by ECM stiffness during growth and development [14,31,32,33,34,35] can be largely improved by carrying out investigations in animal models characterized by a rapid spatiotemporal and easily assessable development.

Monitoring the different developmental stages (from hatching to adult) in the leech *Hirudo verbana* has allowed us to provide a global and extensive picture of the pattern of ECM changes (in terms of both components and spatial organization) in relation to the migration, polarity and differentiation of embedded cells that form the body wall.

A key difference between adult and new-born leeches is their size, since adult leeches are about ten times bigger than new-born animals. The latter, once hatched outside the cocoon, are dimensionally very small and, even if they prefer to live attached to the substrate, are mobile and able to swim. After 50 days the juvenile leeches show further increased motility, and their active swimming is comparable to that of the adult leech.

In spite of the morphological similarities, the body wall organization undergoes abrupt changes from hatching to adult in terms of thickness, number of muscle cells, degree of differentiation and, above all, the different organization of the ECM.

More precisely, the thickness of the body wall increases from about 100–150 μm in the hatching up to 800–1000 μm in the adult, due to a massive numerical and dimensional increase in muscle fibres. During development, such an increase in muscle mass is inversely proportional to the concomitant decrease in structured matrix production. In a radial direction (from the centre toward the body surface) the spatial distribution of muscles was found to dramatically change according to the level of ECM stiffness, which in turn correlated to leech developmental stages. In new-born leeches, a dense connective tissue represents the largest portion of the animal body wall, with less differentiated muscle fibres embedded in connective tissue predominating in the innermost part of the body, whereas very small well-developed fibres are found close to the epithelium. The main features of these muscle fibres are the reduced contractile material and their irregular profile, typical of a migrating phenotype. As leeches grow, larger, differentiated, motile elements with complementary surfaces form groups that are more and more organized closer to the external side. This evidence suggests that the body wall’s growth proceeds thanks to the centrifugal migration of myocytes during active maturation. In juvenile leeches, connective tissue predominates over muscular tissue while, in adults, the connective tissue becomes scarcely represented, as shown by both immuno- and histo-chemical data and quantitative analysis of collagens.

In addition, the number of differentiating cells drastically increases starting from new-born to adult leeches.

As far as the ECM is concerned, its structural organization, as well as the amount of fibrillar and non-fibrillar collagen (the major component of ECM), were found to be deeply different from hatching leeches to adult ones, as shown by both TEM and SEM analysis.

In adult leeches, a loose fibrillar material is localized in a random pattern around each muscle fibre, while a poorly represented thick and compact collagen network embraces each group of muscles, thus defining a connective capsule. A few collagen fibres are embedded in a loose parenchyma under the epithelium and in the space surrounding the gut. At this stage, collagen fibres are typically anisotropic and curly. Overall, the amount of fibrillar collagen in adult leeches is rather small [22].

By contrast, during development of new-born leeches, collagen fibrils are arranged hierarchically, by forming structures progressively thicker and more linearized. The linearized collagen fibres are stiffer than curly ones and, as a result, bundles of parallel elements spatially organized in a lattice are produced, where hexagonally packed collagen fibrils are interconnected by proteoglycan bridges [22,23]. In vertebrates, similar conditions generate mechanical forces on fibroblasts and contribute to sustain the expression of several ECM components, resulting in a matrix spatial pattern, which carries collagen fibrils aligned in a specific direction [35]. Furthermore, it is well known that mechanical forces affect not only the balance between ECM synthesis and degradation, but also the behaviour of differentiating cells [12,36,37].

Of note, these supporting collagenic “struts” (pylons), characterised by their own supramolecular organization, assemble into a solid extracellular network to efficiently fulfil the requirement for developmental processes in leeches.

The collagen scaffold separates and defines radially oriented regions (characterised by loose interstitial matrix) that can be thought of as “highways” to foster migrating cells. Furthermore, it drives the migration of muscle fibres and immunocytes from the centre to the body surface, while at the same time counteracting the pulling and distension forces produced by tensile cell movements (i.e., cell migration speed obeys the concentration of the ECM substrate).

ECM is mainly composed of collagen I, III, and IV, but the percentage of each type changes from hatching to adult. Collagen I remains the main component, but it is present in a larger quantity in hatched and juvenile leeches compared to adults. Collagen III is co-expressed with type I but its amount progressively decreases with ongoing development. This observation is consistent with the crucial role that this type of collagen plays in efficient collagen I fibrillogenesis [38], by regulating the density and the alignment of collagen fibres and myofibroblasts, thus ensuring tissue functionality [17,39,40,41]. Likewise, collagen IV, that provides other macromolecules with a scaffold for the basal membrane, is highly expressed during the first steps of leeches’ development. The regulation of this collagen type is consistent with its function to provide structural integrity and functionality of basement membranes under conditions of increasing mechanical demands and at the same time to regulate cell adhesion and migration. In other invertebrates, such as *Drosophila* [8,9,42,43,44] and *Caenorhabditis,* reduced expression of the collagen IV-related gene *Dcg* results in failure of basement membrane formation, triggering embryonic lethality due to the lack of functional muscle attachment sites [42].

Even though the overall ECM architecture is affected by collagen concentrations, other components play important roles by being expressed in a timely manner. Proteoglycans are highly expressed in new-borns and juveniles (as demonstrated by Alcian blue staining), according to their function of bridging collagen fibrils and keeping them aligned, as well as maintaining the laterally ordered arrays of the newly formed fibrils, as shown in ultrastructural images.

Proteoglycans are made of a core protein with different attached GAG chains, among which hyaluronan (HA) plays a crucial role [2,43]. Noteworthy, the HA expression pattern observed during the first phases of leech development is consistent with its ability to form aggregates and promote the assembly of collagen molecules into fibrils (thereby stiffening the tissue stroma) as well as to regulate cell behaviour. GAGs and HA not only promote invasion and migration of cells, but also play an important role in the formation of pericellular matrix.

As such, during leeches’ development the ECM changes in structure and composition, leading to a modification of the microenvironment where different mechanical cues impact on cells’ behaviour (Figure 10).

How cells perceive and convert external mechanical forces into intracellular signals has been studied extensively [12,44,45,46]. It is well known that, through the process of mechanotransduction (i.e., the mechanism by which cells convert mechanical stimuli into biochemical signals), modulation of ECM stiffness has a broad impact on cell behaviour [14,35,36,37,47,48,49].

In leeches, ECM stiffness promotes and potentiates the movements of myoblasts/myocytes and immunocytes that show typical features of migrating cells, with a surrounding thick basal membrane and extending cytoplasmic processes. In particular, myocytes, in recently hatched and juvenile leeches rather than in adults, show a different degree of development from the centre towards the body surface, while the complementarity of muscle cell profiles become more evident in adult leeches. One of the most relevant findings that emerged from our studies on *H. verbana* development is not only the short time-acting (fifty days) and highly dynamic role of ECM in a wide range of developmental process such as migration, differentiation, and spatial organization of cells), but also the strong reciprocal crosstalk between ECM and cells, which can in turn direct gene expression patterns and, ultimately, cell fates.

Over the past decade, many researchers have focused on how cells perceive and respond to changes in ECM composition and mechanics [37,48,49,50,51,52]. In particular, an impressive bulk of data in the literature has been focused on YAP/TAZ as mediators for converting mechanical signals into transcriptional products [30,35,51,53]. The localization pattern of these proteins within the cell is tuned by different external mechanical stimuli. Within a microenvironment characterized by high ECM stiffness, YAP/TAZ translocate and accumulate in the nucleus to assure the maintenance of stemness, whereas in the context of an ECM with loose properties, YAP/TAZ are retained in the cytoplasm and promote cell differentiation [48,49,53,54].

During leech development, cells are also able to sense and readily respond to mechanical stimuli due to YAP1′s different effects, strictly dependent on the timing of its expression and activation. It is important to note that until now, YAP1 has been identified in *C. elegans* (despite the high level of sequence divergence) [55], and *Drosophila melanogaster* [8] but no specific data regarding Annelida has been reported so far.

We thus report for the first time, the occurrence of a putative YAP1 homolog in leeches. The functional role of leech YAP1 has been confirmed by immunolocalization: muscle cells surrounded by an ECM with loose properties (as validated by ultrastructural images) show a cytoplasmic YAP1 signal, whereas cells in a stiffer microenvironment display a nuclear signal.

Overall, our data strongly suggest the pivotal role of ECM during development and morphogenesis in *H. verbana*. During different leech developmental stages, ECM remodelling leads to a gradient of stiffness that regulates not only the path of migratory cells, but also cell fates within the developmental body wall that respond to changes in the ECM 3D network by activating epigenetic machinery, whose main effectors in the Hippo pathway are YAP1 and TAZ. It is important to underline that *H. verbana* YAP1 can be aligned to *H. vulgaris*, *T. spiralis* and *M. galloprovincialis* amino acid sequences, thus proving the high conservation of a key WW motif in the medicinal leech protein. Similar results were also obtained by comparing human and leech YAP1, revealing a fundamental similarity with vertebrates as well, and confirming the relevant and evolutionarily conserved role of this pathway.

## 4. Materials and Methods

All experiments were performed in three independent replicates. All chemicals were purchased from Sigma–Aldrich (Saint Louis, MO, USA), unless otherwise indicated.

### 4.1. Leech Maintenance and Dissections

All animals (*Hirudo verbana*, Annelida, Hirudinea) were bought from Ricarimpex, Eysines, France and housed in aerated tanks, in an incubator at 20 °C. The cocoons were kept on humid moss, while the adult and the juvenile leeches (50 days after hatching) were kept in lightly salted water (NaCl 1.5 g/L in distilled water). Animals were anesthetized by immersion in a 10% ethanol solution and then dissected to remove body wall tissues.

All collected samples were processed for the experimental uses reported in the following paragraphs.

### 4.2. Light Microscopy and Transmission (TEM) and Scanning (SEM) Electron Microscopy

Samples were collected and fixed with 4% glutaraldehyde in 0.1 M cacodylate buffer (pH 7.4) for 2–4 h at room temperature and then washed for 10 min, three times in the same buffer.

For TEM analyses, specimens were post fixed for 1 h in 1% osmic acid in cacodylate buffer. After dehydration in an ethanol series, samples were embedded in an Epon–Araldite 812 mixture and sectioned with a Reichert Ultracut S ultratome (Leica, Nussloch, Germany). Semithin sections (70 μm) were stained by crystal violet and basic fuchsin and then observed with a light microscope (Eclipse Nikon, Amsterdam, Netherlands); images were acquired with a Nikon DS-SM camera. Thin sections (70 nm) were stained by uranyl acetate and lead citrate and observed with a Jeol 1010 electron microscope (Jeol, Tokyo, Japan).

For SEM analysis, specimens were post-fixed for different times in a solution of 1% osmium tetroxide and 1.25% potassium ferrous-cyanide and dehydrated in an ethanol series and two times in hexamethyldisilazane. The samples were mounted on carbonated stubs and gold coated in an Emitech K250 sputter coater (Emitech, Baltimore, MD, USA) and observed with a SEM-FEG XL-30 microscope (Philips, Eindhoven, The Netherlands).

### 4.3. Histochemistry at Electron TEM: Alcian Blue Staining

In order to stain proteoglycans, samples were treated according to Ruggeri et al. [56]. Tissues were fixed for 2–4 h at 4 °C in 3% glutaraldehyde in 0.1 M PBS (pH 7.4) and then reduced into very small blocks (about 100 μm). After 1 h of preincubation at room temperature in 25 mM acetate buffer (pH 5.8, containing MgCl_2_ at the same concentration used in the subsequent staining solution), samples were incubated in the staining solution (0.05% *w/v* Alcian Blue 8 GX in 25 mM acetate buffer and MgCl_2_ at final concentrations of 0.05 M, 0.3 M, 0.8 M, 1 M, 1.2 M) for 7 h. After rinsing for 1 h in MgCl_2_–acetate buffer, samples were placed first in 0.01 N HCl for 4 h, then in distilled water for 1 h, and finally in PBS for 30 min. After dehydration in an ethanol series, the tissue blocks were embedded in Epon resin and sectioned as previously described.

### 4.4. Masson’s Trichrome, Sirius Red and Alcian Blue Staining

Samples have been fixed in 4% paraformaldehyde for 2 h and then washed three times in PBS buffer. Subsequently, tissues were dehydrated in an increasing scale of ethanol, and paraffin embedded. Sections obtained with a paraffin microtome (7 μm thick) were processed for:-Masson trichrome (Masson trichrome kit, Bio Optica, Milano, Italy) and Sirius red staining (Sirius red picrate, Bio Optica) were performed with ready-to-use kits, according to the datasheets.-Alcian blue 8 G pH 2.5 staining. Briefly, sections were incubated for 30 min at room temperature in the staining solution (0.3% *w/v* Alcian Blue 8 GX in 3% acetic acid pH 2.5) and then differentiated with 3% acetic acid solution for 10 min. After washing, samples were treated with haematoxylin for 2 min, to counterstain nuclei.

Images were recorded with an Eclipse Nikon microscope (Nikon, Tokyo, Japan). The images were acquired using the objective 10× and 20× for the adult stage, and 50× and 100× for the juvenile and the hatching stages.

### 4.5. Immunofluorescence

All steps were performed at room temperature. Cryosections were rehydrated with PBS (pH 7.4) for 10 min and then pre-incubated for 30 min in blocking solution (2% Bovine Serum Albumin and 0.1% Tween20 in PBS, also used to dilute both the primary and the secondary antibodies). Samples were then incubated for 90 min with the primary antibodies (Table 1) and after several washes in PBS buffer, they were incubated for 1 h with suitable secondary antibodies conjugated with cyanin 3 (Cy3, Abcam, dilution 1:400, Cambridge, UK). Nuclei were counterstained with 4′,6-diamidino-2-phenylindole (DAPI, 0.1 mg/mL in PBS) for 5 min and slides were mounted with Cityfluor (Cityfluor Ltd., London, UK). Negative control experiments were performed omitting primary antibodies. Slides were finally observed under a fluorescence microscope (Eclipse Nikon) equipped with the emission filters 360/420 nm for DAPI nuclear staining and 550/580 nm for CY3 signals. Images were recorded with a Nikon digital sight DS-SM (Nikon, Tokyo, Japan) and mounted with Adobe Photoshop (Adobe Systems, San Jose, CA, USA).

Immunolocalization assays for HA (hyaluronic acid) were carried out using a biotin-labelled HA-binding protein (dilution 1:200), which recognizes HA saccharidic sequences. Sections were incubated with biotin-labelled HABP in blocking solution, overnight at 4 °C and, after washing, they were incubated for 1 h with streptavidin FITC-conjugated antibody (dilution 1:250).

### 4.6. Western Blot

Tissues obtained from hatching, juvenile and adult leeches were immediately frozen in liquid nitrogen and then homogenized with a T10 basic ULTRA-TURRAX (IKA, Staufen, Germany) in 10 mL of RIPA buffer (50 mM of NaCl, 1% NP-40, 0.5% sodium deoxycholate, 0.1% SDS, 50 mM of Tris–HCl, pH 7.5, protease/phosphatase inhibitors cocktail) per mg of tissue. The lysates were clarified by centrifugation (13,000 rpm at 4 °C for 20 min) and protein concentration was determined with the Bradford method (Serva, Heidelberg, Germany). Protein extracts were subjected to 8% SDS-PAGE (60 or 120 μg protein each lane); separated proteins were transferred onto 0.45 μm pore size nitrocellulose membranes (Amersham Protran Premium, GE Healthcare, Chicago, IL, USA). The filters were blocked for 2 h at room temperature with 5% (*w*/*v*) non-fat dried milk in TBS (Tris-buffered saline) and then incubated for 2 h at room temperature with the primary antibodies (Table 1) diluted in TBS/5% milk; detection of glyceraldehyde 3-phosphate dehydrogenase (GAPDH) was used as loading control. After three washes of 10 min in TBST (Tris-buffered saline containing 0.1% Tween-20), the membranes were incubated for 1 h with horseradish peroxidase conjugated anti-rabbit (dilution 1:7500 in TBS/5% milk; Jackson ImmunoResearch Laboratories, West Grove, PA, USA) secondary antibody. Finally, the membranes were exposed to the enhanced chemiluminescence substrate (LiteAblot PLUS, EuroClone), followed by autoradiography on X-ray film (KODAK Medical X-Ray film, Z&Z Medical, IA, USA). Densitometric analysis was assessed with the ImageJ software package1. The values are reported as the relative optical density of the bands, normalized to GAPDH.

### 4.7. H. verbana Yes-Associated Protein 1 (YAP1) Alignment and Conservation Analyses

The FASTA format of *H. verbana* YAP1 protein sequence was obtained from the Central Nervous System (CNS) leech transcriptome [57] to perform bioinformatic analyses. The presence of the YAP1 specific WW domain was verified by the Expasy ScanProsite tool (https://prosite.expasy.org/scanprosite/; accessed on 3 August 2022). To determine the possible conservation with other invertebrate species, a multiple sequences alignment was performed using Clustal Omega (https://www.ebi.ac.uk/Tools/msa/clustalo/; accessed on 3 August 2022), comparing *H. verbana* YAP1 with the amino acid sequences of *Hydra vulgaris* (CDG70925.1), *Trichinella spiralis* (KRY35109.1) and *Mytilus galloprovincialis* (VDI03718.1) present in the National Center for Biotechnology Information (NCBI) database (https://www.ncbi.nlm.nih.gov/; accessed on 3 August 2022). A single alignment was conducted between *H. verbana* and *Homo sapiens* to attest to the correct recognition of the primary antibody used in the experiments. The graphical outputs were created with the Jalview software (http://www.jalview.org/; accessed on 3 August 2022) [58].

## Figures and Tables

**Figure 1 ijms-23-15953-f001:**
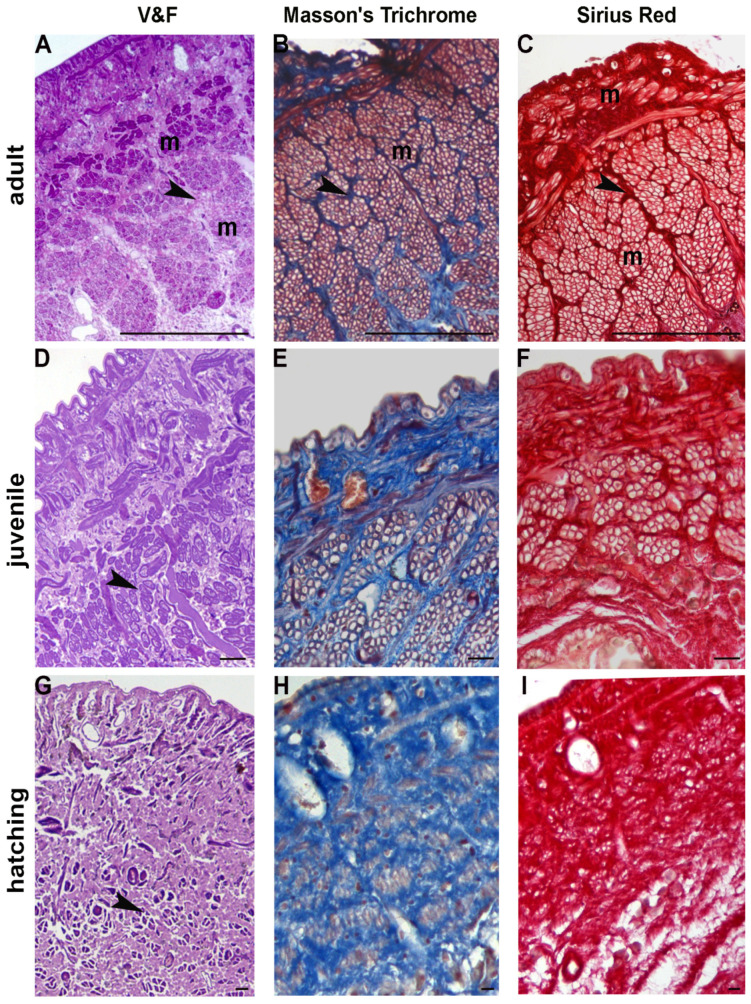
Adult, juvenile and recently hatched *H. verbana* in cross section. Optical microscopy. (**A**–**C**) General view of the body wall of adult *H. verbana* mainly composed of tightly packed muscle fibres (m) wrapped by a monolayered epithelium. Helical muscle fibres (m) are organized in groups separated by loose connective tissue (black arrowhead). The muscular-cutaneous sac is coloured with basic fuchsin and crystal violet (**A**). Masson trichrome (**B**), and Sirius red (**C**) provided a highly detailed and contrasted staining of connective tissue, in blue and red, respectively (black arrowheads). (**D**–**F**) General view of the body wall of *H. verbana* 50 days from hatching. Muscle fibres (black arrowhead) are small, and the connective tissue is predominant over muscle tissue. The muscular-cutaneous sac is coloured with basic fuchsin and crystal violet (**D**). Masson trichrome (in blue; (**E**)), and Sirius red (in red; (**F**)) confirmed the presence of abundant connective tissue. (**G**–**I**) General view of the body wall of recently hatched *H. verbana*. Muscle fibres (black arrowhead) are few and very small (**G**). Both Masson (**H**) and Sirius Red (**I**) allow an evaluation of the massive presence of collagen. Note the leap in scale in measurements. Scale bars: (**A**–**C**) 50 μm; (**D**–**I**) 10 μm.

**Figure 2 ijms-23-15953-f002:**
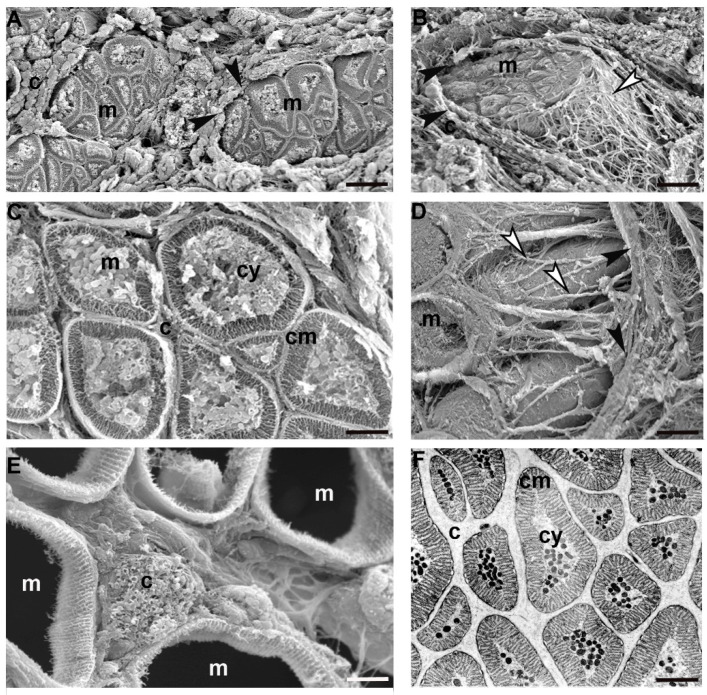
Adult *H. verbana* in cross section. Scanning (SEM) and transmission (TEM) electron microscopy. SEM (**A**–**E**) and TEM (**F**) analysis. Circomyarian muscle fibres (m) formed groups lined by connective tissue (black arrowheads). Collagen scaffold (white arrowheads), surrounding each group of fibres, ensure a mechanical link among muscle fibres and connective capsule. (**B**,**D**). Fibres, characterized by a ring of contractile material (cm) encircling a cytoplasm core (cy), are separated from their neighbours by non-structured extracellular matrix (c) better visible with SEM radical fixation (**F**). Scale bars: (**B**) 12.6 μm; (**A**) 12 μm; (**C**,**D**) 4.5 μm; (**F**) 3.4 μm; (**E**) 1.3 μm.

**Figure 3 ijms-23-15953-f003:**
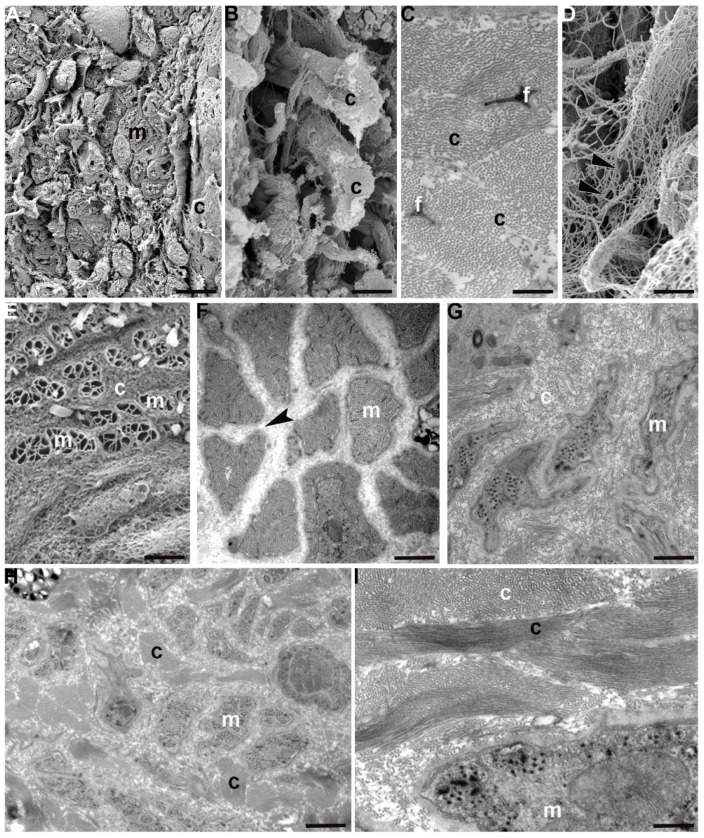
Juvenile *H. verbana* (50 days from hatching) in cross section. Scanning (SEM) and transmission (TEM) electron microscopy. SEM (**A**,**B**,**D**,**E**) and TEM (**C**,**F**–**I**) analysis of cross sectioned leech. Under the epithelium, muscle fibres (m) embedded in a structured ECM characterized by bundles of fibrillar collagen (c) are visible (**A**). (**B**) Detail of the collagenic scaffold. Collagen fibrils are regularly arranged in parallel to form bundles (c). (**C**) TEM micrograph of cross-sectioned collagenic bundles. Each bundle is made of fibrils that are hexagonally arranged and tightly packed. Fibroblast tapered tips (f), showing membrane expansions (giving to the cells a stellate shape), embrace collagen fibrils (c). (**D**) ECM close to the cells show a loose fibrillar network. (**E**) In the proximity of the body surface, large amounts of collagenic scaffold (c) and well developed and grouped muscle fibres (m) are visible. Muscle fibres are very small in size and due to the peculiar SEM preparation, only their membranes are evidenced (m). (**F**) TEM micrograph of differentiated muscle fibres (m) showing the contractile material organized in sarcomeres. In each group, muscle fibres are characterized by complementary adjacent surfaces (black arrowhead). An ECM with loose properties is interposed among fibres. (**G**,**H**) Detail of the innermost part of cross sectioned body. Bundles of collagen (c), arranged radially, delimits the corridors occupied by an ECM with loose properties in which small migrating myocytes (m), less differentiated in respect to the external ones, are visible. (**I**) Detail of myocytes in the migrating phenotype (m) showing the exiguous number of myosin and actin filaments. Scale bars: (**E**) 15 μm; (**A**) 6 μm; (**D**) 3.2 μm; (**B**) 2.2 μm; (**F**) 700 nm; (**H**) 600 nm; (**G**) 500 nm; (**C**,**I**) 300 nm.

**Figure 4 ijms-23-15953-f004:**
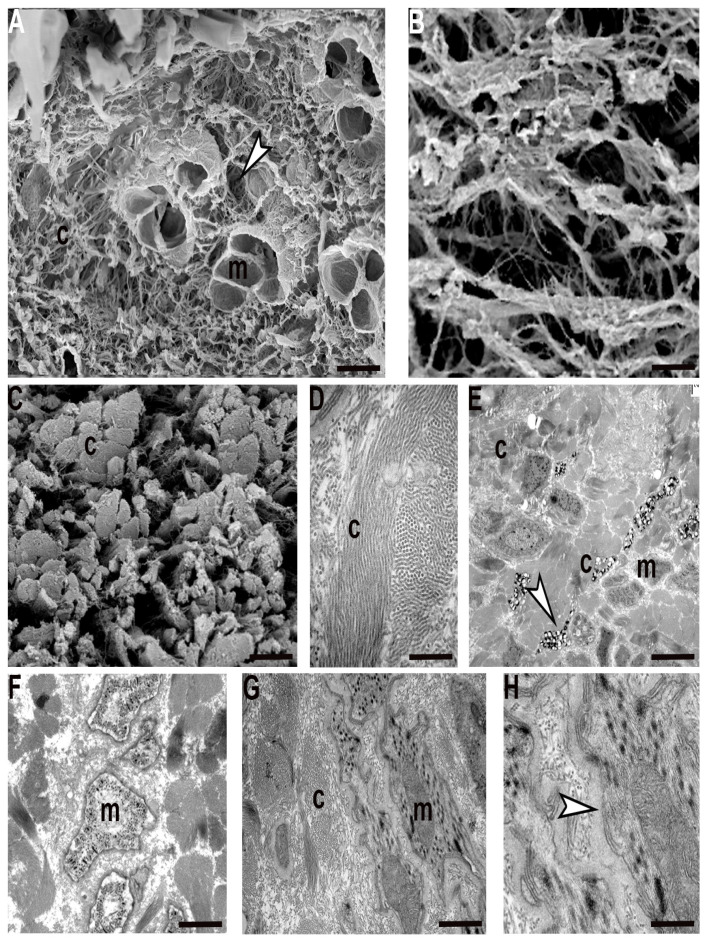
Recently hatched *H. verbana*, in cross section. Scanning (SEM) and Transmission (TEM) electron microscopy. SEM (**A**–**C**) and TEM (**D**–**H**) analysis of cross sectioned leech. The ECM, and in particular collagens (c), are predominant in respect to the myocytes (m). The muscle fibres, very few and small in size (m), are embedded in the ECM with loose properties (white arrowhead) (**A**). Different spatial and structural organization of collagen in the formation of scaffold, that can be loose (**B**) or stiff due to the presence of collagenic bundles (c), is visible (**C**). (**D**–**H**) TEM micrographs. Thick collagenic bundles (c) (**D**), and loose ECM radially alternated, identify corridors (**E**–**G**) occupied by immunocytes (white arrowhead, (**E**)) and myocytes (m, (**E**,**F**)) with ruffled profiles (**F**,**G**). Detail of migrating myocyte showing a thick basal membrane (white arrowhead, (**H**)). Scale bars: (**A**,**C**) 4.3 μm; (**B**,**E**) 2.5 μm; (**G**) 2 μm; (**F**) 1.7 μm; (**H**) 500 nm; (**D**) 400 nm.

**Figure 5 ijms-23-15953-f005:**
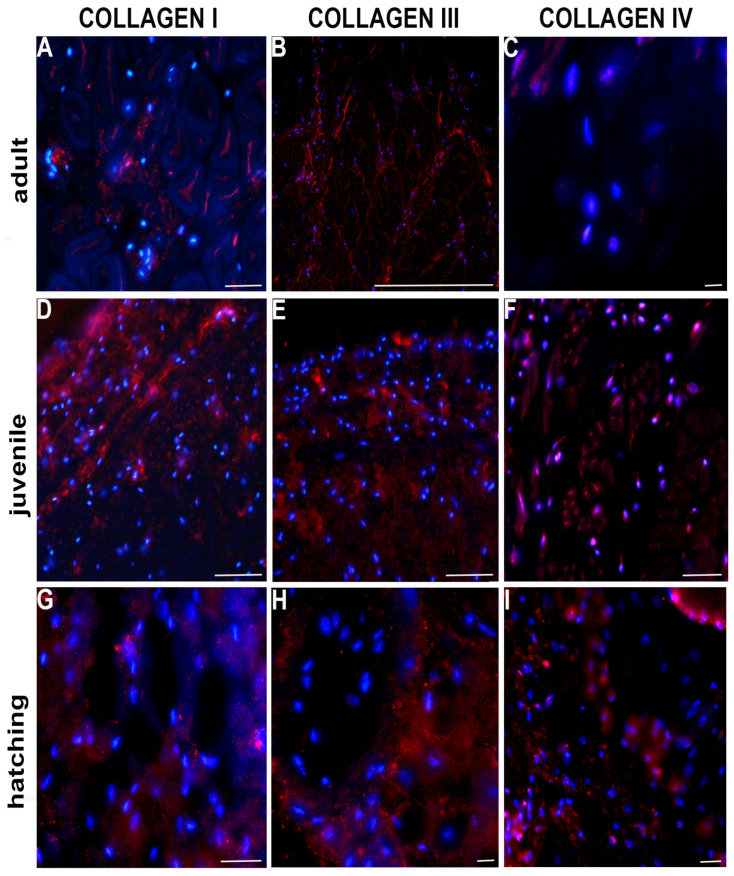
Immuno-characterization of collagen in cross sectioned adult, juvenile and recently hatched *H. verbana.* Immunofluorescent analyses show many cells positively reacting with collagen I (**A**,**D**,**G**), III (**B**,**E**,**H**), and IV (**C**,**F**,**I**) antibodies (in red). The expression of collagen, in general, is higher in hatching (**G**–**I**) and juvenile leeches (**D**–**F**) and decreased in the adult (**A**–**C**). Nuclei are counterstained with DAPI (in blue). Scale bars: (**A**–**I**) 30 μm.

**Figure 6 ijms-23-15953-f006:**
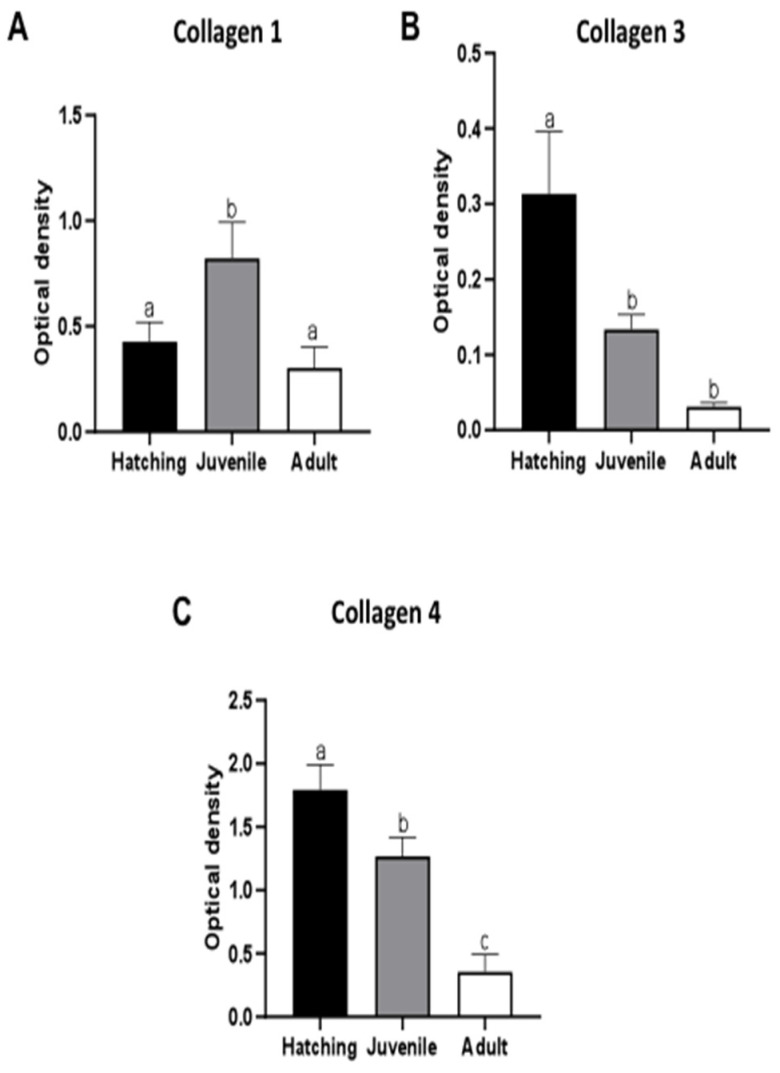
Collagen I, III and IV Western blot analysis in adult, juvenile and recently hatched leeches. Semiquantitative/quantitative analysis on spheroids. The graphs illustrate the expression levels of fibrillar Collagen I (**A**) and III (**B**) and non-fibrillar Collagen IV (**C**). The data result from a densitometric analysis of the Western blots. The values are reported as relative optical density of the bands normalized to glyceraldehyde 3-phosphate dehydrogenase (GAPDH). Statistical differences were calculated by one-way ANOVA followed by Tukey’s post-hoc test; error bars represent SEM and different letters denote statistically significant differences among the stages (*p* < 0.01).

**Figure 7 ijms-23-15953-f007:**
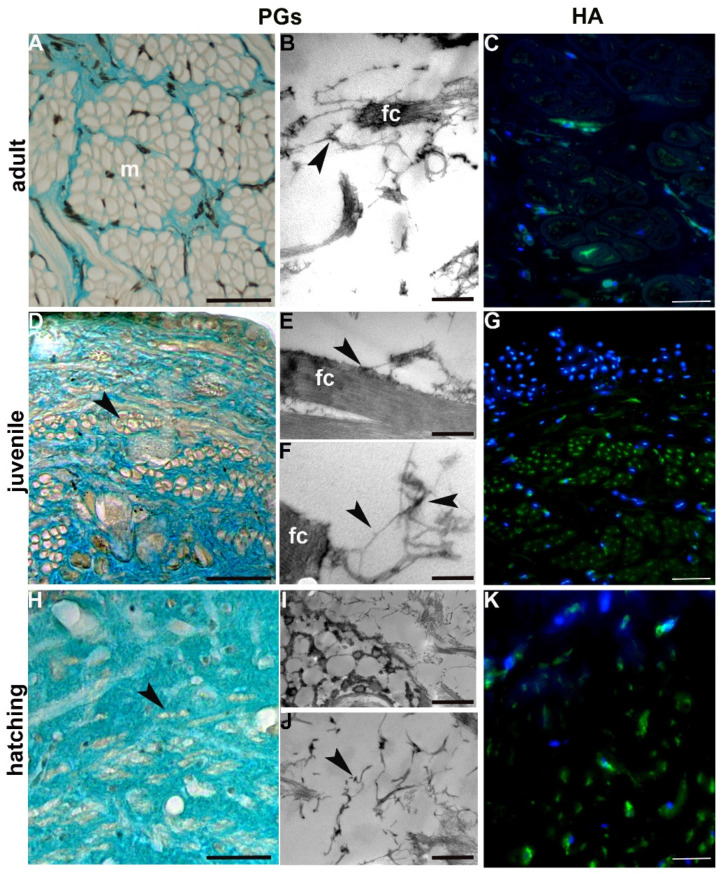
Histochemical, immunocytochemical and ultrastructural analysis of PGs. (**A**,**D**,**H**) Evidence of proteoglycans (green–blue coloured) by Alcian blue histochemical staining; (**B**,**E**,**F**,**I**,**J**) Alcian blue staining (MgCl_2_ concentration = 0.8 M) at TEM. PGs (arrowheads) were visualised as various electron dense filaments closely associated with the collagen fibrils (fc); (**C**,**G**,**K**) Immunostaining with antibodies against HA. Nuclei are counterstained with DAPI (in blue). m, muscle; Scale bars: (**A**) 100 μm; (**D**) 50 μm; (**C**) 40 μm; (**G**) 32 μm; (**H**,**K**) 20 μm; (**I**) 500 nm; (**E**) 400 nm; (**B**) 300 nm; (**F**,**J**) 200 nm.

**Figure 8 ijms-23-15953-f008:**
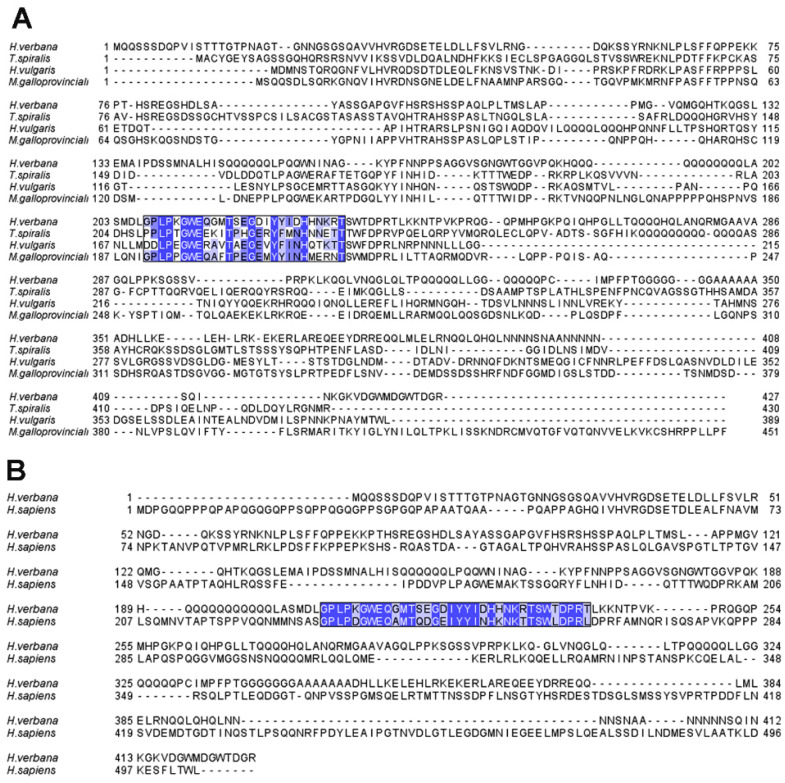
YAP1 conservation analyses. (**A**) The *H. verbana* YAP1 amino acid sequence, identified in the medicinal leech CNS transcriptome has been aligned with those of *H. vulgaris* (Cnidaria, CDG70925.1), *T. spiralis* (Nematoda, KRY35109.1) and *M. galloprovincialis* (Mollusca, VDI03718.1) obtained from the NCBI database. The YAP1 specific WW domain results are highly conserved and different blue shades indicate the diverse conservation of each amino acid. (**B**) The alignment of *H. verbana* and *H. sapiens* YAP1 protein sequences shows the high conservation of the WW domain between these species, validating the correct recognition of the primary antibody used.

**Figure 9 ijms-23-15953-f009:**
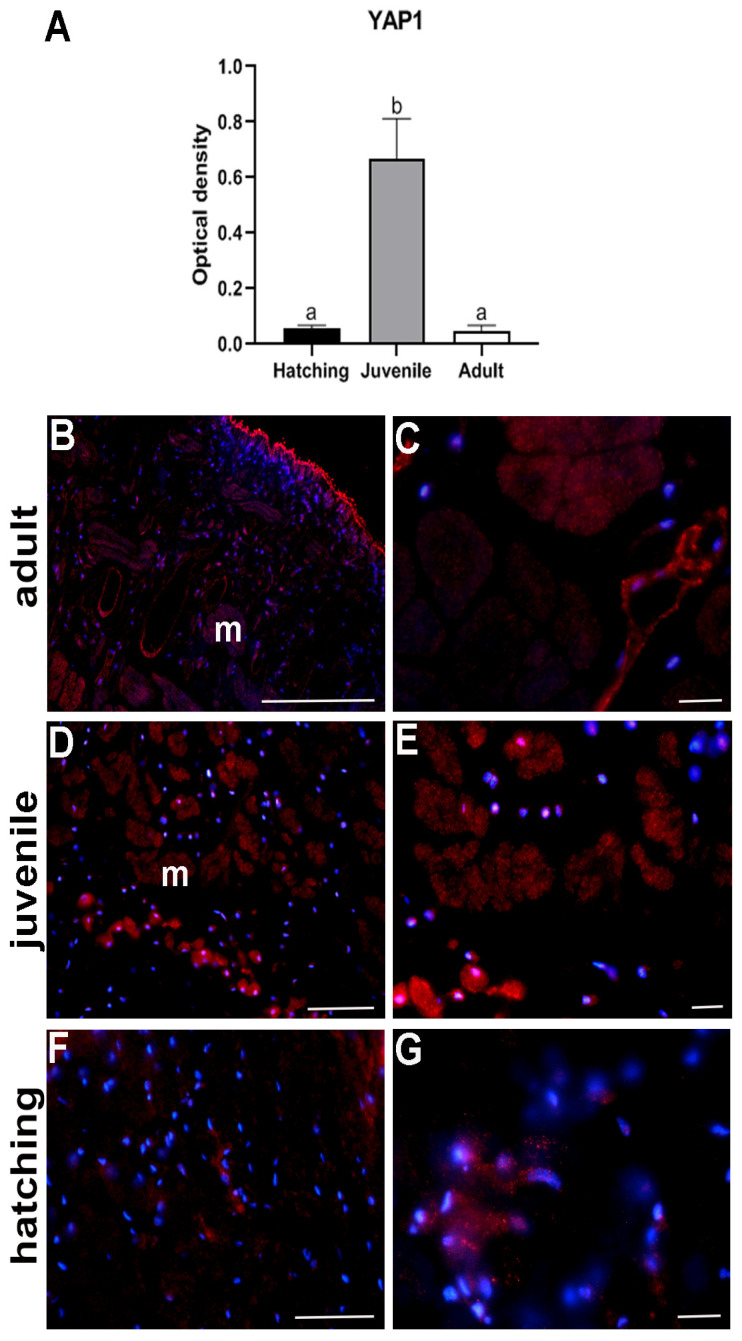
Western blot analysis and immune-characterization of YAP1 in adult, juvenile and hatched leeches. The graph (**A**) shows the expression levels of YAP1 at the three different stages of development. The data result from a densitometric analysis of the Western blots. The values are reported as relative optical density of the bands normalized to glyceraldehyde 3-phosphate dehydrogenase (GAPDH). Statistical differences were calculated by one-way ANOVA followed by Tukey’s post-hoc test; error bars represent SEM and different letters denote statistically significant differences among the stages (*p* < 0.01). Immunolocalization of YAP1 (red signal) confirms the quantitative data and reveals positivity both in cytoplasm and in nuclei of body wall cells (m, muscle), above all in the juvenile (**D**,**E**), compared to adult (**B**,**C**) and hatching leeches (**F**,**G**). Nuclei are counterstained with DAPI (in blue). Scale bars: (**B**) 50 µm; (**C**–**G**) 20 µm.

**Figure 10 ijms-23-15953-f010:**
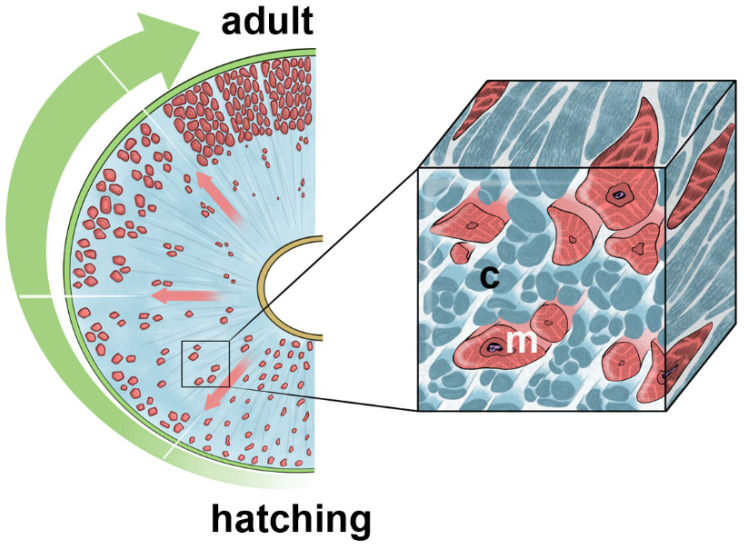
Schematic drawing highlighting modification of the body wall in *H. verbana* during development from hatching to adult. Collagenic scaffold (c) in pale blue and myocytes (m) in pink.

**Table 1 ijms-23-15953-t001:** List of primary antibodies used for immunocytochemical (IF) and immunoblotting (WB) studies.

Antibody	Description	Company	Application	Dilution
Collagen Iα2	Rabbit polyclonal	Sigma-Aldrich (St. Louis, MO, USA)	IF	1:200
			WB	1:500
Collagen IIIα1	Rabbit polyclonal	Proteintech (Rosemont, IL, USA)	IF	1:100
			WB	1:500
Collagen IV	Rabbit polyclonal	Sigma-Aldrich (St. Louis, MO, USA)	IF	1:100
			WB	1:500
HABP	Biotin-labelled	Seikagaku Co. (Tokyo, Japan)	IF	1:200
YAP1	Rabbit polyclonal	GeneTex (Irvine, CA, USA)	IF	1:150
			WB	1:500
GAPDH	Rabbit polyclonal	Proteintech (Rosemont, IL, USA)	WB	1:7000

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
