# Peer review of "Spatio-Temporal Changes of Extracellular Matrix (ECM) Stiffness in the Development of the Leech Hirudo verbana"

_ijms, 2022, doi:10.3390/ijms232415953_

Round 1
Reviewer 1 Report
The manuscript is of interest i the field, and the data are supporting their discussion. However, better organisation of the panel, and the results must be performed.

Reviewer 2 Report
The manuscript reports on the spatio-temporal variations of ECM during the development of LEECH Hidrudo verbena. It presents the influence of 3D ECM change on the fates of cells and tissues and their organisation. It is followed by the analysis of ECM microenvironment architecture from the physical and chemical points of view leading to the adaptation of a specific tissue in form of its inner organisation. Also, the effects of ECM stiffness on the polarity and differentiation of various cells in the course of development are described. The findings help to clarify similar phenomena in human growth and development thanks to a rapid spatiotemporal development. Different techniques were used in the study to perform the aforementioned analysis such as histochemistry, immunofluorescence, and scanning electron microscopy.
The manuscript is well-structured and easy to follow. It is suggested to modify the title so that the stiffness variation of ECM is not considered as the origin of the complex changes in the development course.
In line 270, 530, and 540 the citation format is different from the rest of the manuscript
In Fig. 1 and 5, it would have been much easier for the reader to make a comparison between different cases if the scale bars are of the same dimension.
Reviewer 3 Report
The PGs visualization without special electron-microscopic immunogold markers is not possible. So, concerning these electronograms it is possible only speculate about PGs presence, especially HA. Most likely, one can expect the presence of chondroitin sulfate proteoglycan, named decorin, which "decorates" collagen fibrils, but this is a very small molecule. Your speculation concerning PGs with TEM demandes using of special markers.

Reviewer 4 Report
The manuscript of Pulze et al. is aimed at investigating the interactions between the extracellular matrix (ECM) and cells of the tissue microenvironment (TME) during growth using the model system of the invertebrate Hirudo verbena. Authors employ leech at different developmental stage and evaluate them using a variety of morphological, biochemical and bioinformatic methods (electron microscopy, immune fluorescence, histological staining, Western blot and protein sequence analysis). Based upon the data obtained during their investigations, Authors conclude that ECM stiffness is a critical driving force during developmental transformation serving as a guide for cell migration paths. In addition, ECM alterations also affect cell fate through the activation of epigenetic machinery and that this mechanism is mediated by the specific proteins (YAP1 and TAZ) of the Hippo pathway. Authors complement the manuscript with 9 multipanel Figures, a summary illustration Figure and 1 Table. Authors cite 57 publications to put their findings into context with current literature.
The experiments are described in a detailed fashion and the results support the conclusions of the Authors.
The manuscript fits the scope of the “International Journal of Molecular Sciences” and is of interest for the readers of the journal. The text of the manuscript is written in a clear fashion, the logic of the text is easy to follow. The model organism (leech) employed is appropriate. The figure legends are clearly explained.
This reviewer only noted some minor stylistic issues that need to be addressed before publication.
1. Title: LEECH should be changed to “leech”
2. Title: ECM should be changed to “extracellular matrix (ECM)”
3. Abstract: Page 1, Line 17 “our results clearly show” should be replaced by “our results show”
4. Abstract: Page 1, Line 23: YAP1 should be spelt out at the first mentioning “Yes1 Associated Transcriptional Regulator (YAP1)”
5. Page 4, Line 117: “muscle fibers were small” should be replaced by “muscle fibers are small”.
6. Page 7, Line 190 “to mark the limits of” should not be in bold
7. Page 8, Line 209: Figure 4F: “Just like muscular cells, immunocytes and interstitial cells are also produced in the central zone close to the gut and then migrate along the channels delimited by collagen toward the superficial area (Figure 4F).”
Please mark on the image which are the immunocytes and interstitial cells.
Round 2
Reviewer 1 Report
I'm not satisfied with the feedback to my comments, and many of them were't answered at all in appropriate ay.